# Association of Mindfulness with Perfectionism, Exercise Self-Efficacy, and Competitive State Anxiety in Injured Athletes Returning to Sports

**DOI:** 10.3390/healthcare11202703

**Published:** 2023-10-10

**Authors:** Liang Li, Longjun Jing, Yang Liu, Yiwei Tang, Huilin Wang, Jingyu Yang

**Affiliations:** 1School of Physical Education, Hunan University of Science and Technology, Xiangtan 411201, China; 2China Athletics College, Beijing Sport University, Beijing 100061, China; 3School of Business, Hunan University of Science and Technology, Xiangtan 411201, China; 4School of Social and Political Science, University of Edinburgh, Edinburgh EH8 9LD, UK; 5Department of Medical Bioinformatics, University of Göttingen, 37077 Göttingen, Germany

**Keywords:** mindfulness, perfectionism, competitive state anxiety, exercise self-efficacy, injured athletes returning to sports

## Abstract

Injured athletes often face performance challenges upon returning to the field, influenced by external factors and negative emotions. This study investigates how mindfulness is associated with perfectionism, exercise self-efficacy, and competitive state anxiety in athletes recovering from injuries. Using snowball sampling and convenience sampling methods with a cross-sectional dataset of 359 participants from southern China (collected between October and November 2022), we employed structural equation modelling to analyse the relationship between mindfulness and competitive state anxiety in returning athletes. The results reveal that mindfulness interventions enhance exercise self-efficacy, boost task-related confidence, reshape perfectionism towards a positive outlook, and decrease competitive state anxiety. This study establishes positive correlations between perfectionism and competitive state anxiety, and a negative correlation between exercise self-efficacy and competitive state anxiety. Moreover, exercise self-efficacy and perfectionism partially mediate mindfulness’s positive impact on competitive state anxiety. In conclusion, this research highlights mindfulness’s potential to alleviate perfectionism and competitive state anxiety while enhancing exercise self-efficacy among athletes on the road to recovery.

## 1. Introduction

Anxiety affects various aspects of human psychology and physiology, making it one of the most controversial and significant topics in psychology [1]. It is a social psychological factor that often manifests as bodily tension and worry or unease about something. Anxiety can be classified into state anxiety and trait anxiety [2], with state anxiety being a transient state dependent on a specific situation or event, while trait anxiety is a relatively stable personality characteristic. Athletes, who represent their teams or countries in sports events, undergo professional training to showcase their athletic abilities during competitions. Emotions, including anxiety, are known to accompany athletes during competitions, posing risks of injury [3], particularly as the level of competition difficulty increases. Furthermore, as the level of competition difficulty increases, it raises anxiety levels. Athletes assess the difficulty of the task and evaluate their own ability to accomplish it, which involves an analysis process that can lead to negative emotions such as anxiety and fear [4].

Studies have shown that injuries among group gymnasts are primarily caused by the difficulty of the required tasks, resulting in high levels of anxiety among athletes. Additionally, after recovering from an injury, athletes may develop psychological barriers and refuse to perform technical movements they believe could lead to re-injury [5]. Moreover, upon returning to the field after recovery, athletes may experience excessive concerns about meeting the expectations of their coaches, family, teammates, and audience, as well as the fear of not performing at the same level as before the injury or losing to their opponents. Therefore, athletes with high self-esteem who overly focus on their own performance may experience higher levels of anxiety.

Psychological training in the field of sports aims to help athletes cope better with challenges during competitions. Psychological skills training (PST) and mindfulness are two mainstream forms of psychological training, each with a different theoretical background [6]. PST provides athletes with skills such as goal setting, imagery, self-talk, and relaxation to enhance attention and control negative psychological processes that may affect performance [7]. On the other hand, mindfulness, unlike PST, is a form of meditation that involves consciously and non-judgmentally being aware of present experiences, including bodily sensations and emotional states [8]. It encourages individuals to face current experiences with openness and acceptance, redirecting attention from the pressures arising from technique, physicality, and professionalism. Some studies have indicated that mindfulness is considered the most beneficial intervention in alleviating anxiety [9,10], helping athletes develop their ability to cope with negative emotions and enhance performance [11,12]. Currently, mindfulness in sports is applied through various methods, including mindfulness acceptance commitment (MAC) [13], mindfulness sport performance enhancement (MSPE) [14], and mindfulness meditation training for sport (MMTS) [15].

Existing research on competitive state anxiety among athletes post-recovery typically focuses on the impact of anxiety itself, lacking in-depth exploration of competitive state anxiety. Studies have shown that athletes with higher levels of competitive anxiety are more prone to injury [16]. This study posits a close relationship between perfectionism and competitive state anxiety. Perfectionism, a multidimensional trait, can be categorized into positive and negative perfectionism. Negative perfectionism is unhealthy, characterized by high levels of perfectionist efforts and concerns. Athletes with negative perfectionism engage in critical self-evaluation, excessively focus on discrepancies between expectations and outcomes, and worry about errors and others’ expectations, often leading to negative emotions, low self-esteem, and reduced self-efficacy [17].

Mindfulness originated from Eastern Buddhism and has a history of one thousand years. Through mindfulness practice, the athlete’s mental state and psychological emotions can be altered, alleviating physical pain, reducing anxiety, and enhancing attention [18,19,20]. Research has shown a connection between mindfulness and exercise self-efficacy [21]. By engaging in effective mindfulness practices, athletes can improve their awareness of tasks or environments, helping them increase exercise self-efficacy after an injury [22] and fostering a stronger belief in their abilities to achieve their intended athletic goals. Athletes with high exercise self-efficacy are better able to adjust their state when faced with challenging tasks, rather than doubting their abilities.

It is well known that athletes experience pre-competition anxiety, and the impact of anxiety on their performance is widely recognized [23]. In particular, athletes who have recovered from an injury often exhibit higher levels of competitive state anxiety due to their fear of re-injury and failure compared to uninjured athletes [24]. Mindfulness is an effective approach for reducing competitive state anxiety [25]. Active mindfulness practice enhances athletes’ positive emotions, alleviates pre-competition pressure and anxiety [9,26], strengthens their confidence, and improves their athletic performance. Additionally, exercise self-efficacy plays a significant role in task completion and performance. Research has demonstrated that higher exercise self-efficacy reduces anxiety and increases athletes’ confidence levels and competitive state [27,28], leading to better performance. Conversely, lower exercise self-efficacy is associated with higher levels of anxiety and poorer performance states in athletes [29]. Therefore, this study proposes the following hypotheses:

**Hypothesis 1 (H1):** 
*Mindfulness is positively correlated with exercise self-efficacy.*


**Hypothesis 2 (H2):** 
*Exercise self-efficacy is negatively correlated with competitive state anxiety.*


**Hypothesis 3 (H3):** 
*Mindfulness is negatively correlated with competitive state anxiety.*


Research has shown that negative perfectionism involves critical self-evaluations of one’s own performance, a sense of discrepancy between expectations and outcomes, judgment of mistakes, and concerns about others’ high expectations [30]. Negative perfectionism is associated with various psychological disorders such as depression, obsessive–compulsive disorder, and excessive anxiety [31], and levels of perfectionism among young people have been found to be significantly higher than in the past [17]. In the field of sports, athletes are expected to achieve better performance and set high standards for themselves. However, athletes with a tendency towards negative perfectionism may experience more negative emotions and pre-competition anxiety due to their excessive pursuit of perfection [32]. Mindfulness plays a beneficial role in highly anxious and perfectionistic individuals. As a tool, mindfulness can help post-injury athletes enhance self-awareness, reshape perfectionism, reduce levels of negative perfectionism, and increase levels of positive perfectionism. Therefore, this study proposes the following hypotheses:

**Hypothesis 4 (H4):** 
*Mindfulness is negatively correlated with perfectionism.*


**Hypothesis 5 (H5):** 
*Perfectionism is positively correlated with competitive state anxiety.*


When post-injury athletes engage in training or competition, they perceive a higher risk of re-injury and experience greater fear compared to uninjured athletes, leading to increased pressure. This pressure can cause post-injury athletes to develop fear and apprehension towards re-injury, which can impact their performance [33]. Fear of re-injury can result in psychological barriers or refusal to perform specific technical movements [34]. When post-injury athletes are afraid and perceive potential threats, they experience anxiety. Current research on the mediating role of exercise self-efficacy primarily focuses on self-regulation [10], pre-competition anxiety [35], and state anxiety [26]. Studies have found a relationship between mindfulness, exercise self-efficacy, and athletes’ state anxiety [1,28]. When exercise self-efficacy is stronger, athletes have higher confidence in achieving their goals and experience fewer negative emotions such as fear, worry, and perceived risks. Additionally, research has shown that negative perfectionism can generate negative emotions and lead to feelings of fatigue [34], while mindfulness can adjust negative perfectionism levels, enhance positivity, and consequently reduce competitive state anxiety levels [36]. Therefore, this study proposes the following mediation hypotheses:

**Hypothesis 6 (H6):** 
*Exercise self-efficacy mediates the relationship between mindfulness and competitive state anxiety.*


**Hypothesis 7 (H7):** 
*Perfectionism mediates the relationship between mindfulness and competitive state anxiety.*


In summary, the hypothetical model proposed in this study is illustrated in Figure 1.

Based on this, the objectives of this study are as follows: (1) explore the association of mindfulness with exercise self-efficacy, perfectionism, and competitive state anxiety; (2) propose recommendations for reducing athlete anxiety.

## 2. Materials and Methods

### 2.1. Procedure and Participants

This study utilized a dual approach, employing both snowball sampling and convenience sampling methodologies. Snowball sampling proved invaluable for reaching individuals with the desired characteristics, particularly when direct access was challenging. In this method, initial study participants, comprising high-level athletes (national level and above) who had successfully recovered from injuries, played an active role in recruiting additional participants from their personal networks. This chain-referral process continued until data saturation, ensuring a comprehensive representation of the target population.

Simultaneously, convenience sampling was employed to efficiently select readily available participants. To initiate the sampling process, the researchers contacted directors of high-level sports teams across various provinces in southern China. The directors, acting as intermediaries, distributed survey questionnaires to potential participants, streamlining the recruitment of high-level athletes who met the study criteria.

The survey’s purpose was transparently communicated to the participants, emphasizing the anonymous and exclusive utilization of data for academic research. Data collection occurred from October to November 2022, during which 500 questionnaires were distributed to high-level athletes in southern China. Rigorous quality control measures were applied, leading to the exclusion of invalid responses. By the end of November, a robust dataset comprising 359 valid questionnaires was amassed, achieving a commendable response rate of 71.8%.

Table 1 presents the demographic characteristics of the 359 participating athletes. In terms of age, athletes above the age of 25 accounted for 65% of the total sample. Regarding gender, male athletes accounted for 65.2%, while female athletes accounted for 34.8%. Among the athletes, 54.6% were engaged in ball sports, and 37.3% were involved in track and field events. The majority of the respondents were at the secondary level, accounting for 81.9% of the sample.

### 2.2. Measurements

The researchers measured the level of mindfulness using five items from the mindfulness scale developed by Carlson and Brown [5]. Sample items included “It seems I am ‘running on automatic’ without much awareness of what I’m doing”. The measurement of exercise self-efficacy was carried out using five items from the scale developed by Kocak [37]. Sample items included “I have the motor skills required for my sport discipline”. The measurement of perfectionism was derived from the MPS scale developed by Hewitt and Flett [38]. Sample items included “I have extremely high goals for myself in my sport”. Competitive state anxiety was measured using five items from the competitive state anxiety scale for athletes developed by Smith [39]. Sample items included “I am worried about choking under pressure”. All four scales were measured using a 5-point Likert scale, ranging from 1 (strongly disagree) to 5 (strongly agree).

To adapt the scales to the specific research field and cultural background, the researchers made certain adjustments to the wording of the items. Test–retest reliability of the adjusted scales was ensured by conducting a pilot test with high-level university student athletes at a university in Changsha. The researchers distributed 75 questionnaires using convenience sampling and received 64 valid responses. The results showed that Cronbach’s alpha coefficients were all above 0.8, indicating good internal consistency of the measurement instruments [40].

### 2.3. Data Analysis

This study employed AMOS 26.0 (IBM Corp, Armonk, NY, USA) to build a structural equation model (SEM) to test the proposed hypotheses. Following the recommendations of Anderson and Gerbing [41], a two-step modelling approach was used in this study. Firstly, confirmatory factor analysis (CFA) was conducted to assess the fit between the measurement model and the data. Secondly, regression analysis and path analysis were performed on the structural model. Additionally, the study examined the mediating roles of exercise self-efficacy and perfectionism in the relationship between mindfulness and competitive state anxiety. Furthermore, 5000 bootstrap samples were used to test the mediating effects of exercise self-efficacy and perfectionism. Finally, the model’s validity was assessed by examining the fit indices and path coefficients of the hypothesized model.

Common method variance (CMV) was also examined to address the potential issue of CMV in behavioural research. Firstly, Harman’s single-factor test results indicated that the percentage of variance extracted from the single-factor test was 43.427% (below the classical threshold of 50%), suggesting the absence of CMV [42]. Secondly, this study followed the method proposed by Lindell and Whitney [43], which compares the fit of a CFA single-factor model and a CFA bifactor model. The chi-square value for the single-factor model was 3630.574 with 170 degrees of freedom, while the chi-square value for the bifactor model was 426.802 with 164 degrees of freedom. The ratio of the difference in chi-square values to the difference in degrees of freedom was 533.962, indicating a significant difference in chi-square values between the two models, providing evidence against CMV. Therefore, no correction for CMV was necessary in this study.

## 3. Result

### 3.1. Reliability Analysis

Reliability refers to the internal consistency among the factors [44], which is commonly assessed by calculating Cronbach’s alpha coefficient to determine the internal consistency of a scale. As shown in Table 2, the Cronbach’s alpha coefficients for the scales in this study are all above 0.7, which meets the standard proposed by Hair [45].

### 3.2. Convergent Validity

The extent to which a measure reflects a specific construct is known as convergent validity [46]. From Table 2, it can be observed that the factor loadings of each item in the scale are all above 0.5 (λ > 0.5), indicating satisfactory convergent validity for the constructs.

### 3.3. Discriminant Validity

Discriminant validity refers to the uniqueness of constructs [47]. According to the criteria proposed by Fornell and Larcker [40], discriminant validity can be assessed by comparing the square root of the average variance extracted (AVE) for each construct with the correlations between constructs. From the results shown in Table 3, it can be observed that the correlations between constructs are lower than the square root of the AVE for each construct. Therefore, the discriminant validity of the constructs meets the criteria.

### 3.4. Hypothesis Testing

The researchers found that mindfulness is associated with competitive state anxiety through the mediating factors of exercise self-efficacy and perfectionism. The present study employed bootstrapping to test the existence of mediating effects [48].

The results support Hypothesis 1, indicating a significant positive relationship between mindfulness and exercise self-efficacy (*β* = 0.390, *p* < 0.001). Hypothesis 2 is also supported, showing a significant negative relationship between exercise self-efficacy and competitive state anxiety (*β* = −0.367, *p* < 0.001). Hypothesis 3 is supported as well, indicating a significant negative relationship between mindfulness and competitive state anxiety (*β* = −0.16, *p* < 0.05). Moreover, Hypothesis 4 is supported, indicating a significant negative relationship between mindfulness and perfectionism (*β* = −0.534, *p* < 0.001). Lastly, Hypothesis 5 is supported, showing a significant positive relationship between perfectionism and competitive state anxiety (*β* = 0.371, *p* < 0.001).

The results of the bootstrapping analysis with 5000 samples and a 95% confidence interval are presented in Table 4. The Z-values are greater than 1.96, and the 95% confidence intervals do not include zero, indicating significant indirect effects. Additionally, it is found that mindfulness has an indirect effect on competitive state anxiety through the mediating factors of exercise self-efficacy and perfectionism (standardized indirect effect = −0.341, *p* < 0.001), providing support for Hypotheses 6 and 7.

## 4. Discussion

This study makes several contributions to the analysis of athletes who have recovered from injuries. Firstly, existing research has mainly focused on the fear of re-injury, perceived threat of injury, and anxiety among recovering athletes, with limited investigation into the specific causes of anxiety in this population [49]. The researchers believe that studying the causes of anxiety among recovering athletes can help identify the root of the problem and find potential solutions. Similar to previous research, this study confirms that recovering athletes have a higher desire for victory, surpassing themselves, pursuing excellence, and achieving rankings compared to physically healthy athletes [50]. However, this study further explores and highlights that athletes with perfectionistic traits, when returning to competition after recovery, tend to view their performance as a crucial means of demonstrating their self-worth. These athletes exhibit higher personal standards and excessive focus on errors, which in turn leads to competitive state anxiety. It is worth noting that the majority of previous studies on perfectionism have mainly focused on individuals with psychological disorders such as depression, OCD, and eating disorders, with limited research on athletes. However, when recovering athletes return to the field, they experience higher levels of anxiety compared to physically healthy athletes. They face elevated levels of fear of re-injury, concerns about self-esteem, feelings of failure, and social evaluation that surpass those of the general population. These athletes are also expected by coaches and others to perform better in competitions, which further increases their anxiety level [51]. Additionally, the negative emotions associated with the fear of re-injury also contribute to anxiety in recovering athletes. Based on these findings, employing mindfulness techniques to enhance athletes’ self-awareness and reduce negative perfectionism levels can effectively lower their levels of competitive state anxiety. Mindfulness involves cultivating a non-judgmental awareness and acceptance of the present moment, gradually shifting an individual’s attention away from being actively controlled by negative thoughts and focusing on the task at hand [52]. Mindfulness enhances an individual’s ability to regulate and control their responses to events, thereby alleviating anxiety caused by negative emotions.

Moreover, this study explores the relationship between exercise self-efficacy and competitive state anxiety in recovering athletes. The results demonstrate a significant negative correlation between exercise self-efficacy and competitive state anxiety (see Figure 2). Exercise self-efficacy plays a moderating role in the anxiety levels of recovering athletes. When facing intense competition environments, their confidence in their technical skills and experience helps them overcome the fear of re-injury, reduce their levels of competitive state anxiety, and achieve better performance in competitions. The impact of mindfulness on perfectionism is particularly significant, followed by exercise self-efficacy. As shown in Figure 2, perfectionism and exercise self-efficacy mediate the relationship between mindfulness and competitive state anxiety, with mindfulness and the mediating variables explaining 46.3% of the variance in competitive state anxiety. Furthermore, this study provides a promising pathway for studying competitive state anxiety, focusing on the association between mindfulness and perfectionism and further exploring the impact of perfectionism on competitive state anxiety.

The practical implications of the research highlight the potential of mindfulness training as a pre-competition strategy. This study found a negative correlation between mindfulness and competitive state anxiety. Mindfulness training helps athletes develop higher levels of exercise self-efficacy and reduce negative perfectionism, thereby decreasing competitive state anxiety, especially among recovering athletes. When injured athletes undergo treatment and rehabilitation and return to the field, they may experience negative emotions due to various personal and external factors, which can impact their thoughts and cognition, leading to anxiety, negativity, and lack of concentration during competitions. Recognizing the detrimental effects of negative emotions on athletes, the researchers propose that mindfulness training can alleviate the negative emotions arising from psychological fear and anxiety among recovering athletes, enhance their self-efficacy, and instil greater confidence in accomplishing their goals. Therefore, it is highly necessary to provide mindfulness training to athletes before competitions. To help athletes overcome the influence of anxiety and demonstrate their best performance on the field, coaching teams should have professional mindfulness trainers. Offering athletes professional and scientifically designed mindfulness training can assist them in overcoming the distress of negative emotions, avoiding self-judgment, reducing excessive focus on external circumstances, and directing their attention solely to the task at hand. They can then enjoy the process of striving for perfection without fearing imperfect outcomes. MAC [13,53] and MSPE [14] are two mindfulness intervention methods that can be incorporated into athletes’ training programmes.

Additionally, the national sports authorities should allocate more funding for mindfulness training and increase the availability of psychological training courses, including mindfulness, while expanding the pool of mental training experts. Mindfulness training is effective for athletes, particularly those who have recovered from injuries and are returning to competitions. These athletes often experience higher levels of anxiety compared to uninjured athletes. During the treatment and rehabilitation process, they face more complex external environments, with coaches, family members, spectators, and teammates all potentially contributing to their stress levels. However, the focus should be on monitoring athletes’ emotional well-being, rather than solely assessing their performance after returning to competitions. Having excessively high expectations not only affects athletes’ performance during competitions but also increases their anxiety levels, thereby increasing the risk of re-injury. Therefore, coaching teams should provide mindfulness training to athletes, helping them cope with negative emotions and stress in positive ways. Through mindfulness training, athletes can redirect their attention to fully accepting the present task, rather than using negative thoughts to cope with pressures related to techniques, physical conditions, or external environments.

This study has several limitations. Firstly, it is a cross-sectional study, and future research could employ a longitudinal design with experimental interventions and control groups to more accurately and effectively demonstrate the impact of mindfulness on competitive state anxiety. Secondly, the sample selection is limited as it only includes athletes from southern China and lacks representation from elite athletes. Future studies should aim to include a more diverse sample, considering factors such as athletes’ skill levels, educational backgrounds, and geographical locations to enhance the generalizability of research findings. Thirdly, this study lacks an investigation into and classification of the types of injuries sustained by athletes. Categorizing athletes who have experienced injuries and returned to sports into a singular group is overly broad, limiting the depth of our research. It is recommended that future research endeavours delve into the specific types of injuries athletes sustain. This would enable the development of tailored interventions for addressing distinct psychological issues associated with various injury types.

## 5. Conclusions

This study demonstrates that negative emotions experienced by athletes during the recovery process can lead to competitive state anxiety. By reducing negative emotions and enhancing self-efficacy, anxiety can be reduced. The study highlights that both self-efficacy and perfectionism are important factors influencing competitive state anxiety. Specifically, mindfulness can impact competitive state anxiety through its mediation of self-efficacy and perfectionism. Mindfulness training plays a positive role in reducing perfectionism, enhancing self-efficacy, and decreasing anxiety levels among athletes.

## Figures and Tables

**Figure 1 healthcare-11-02703-f001:**
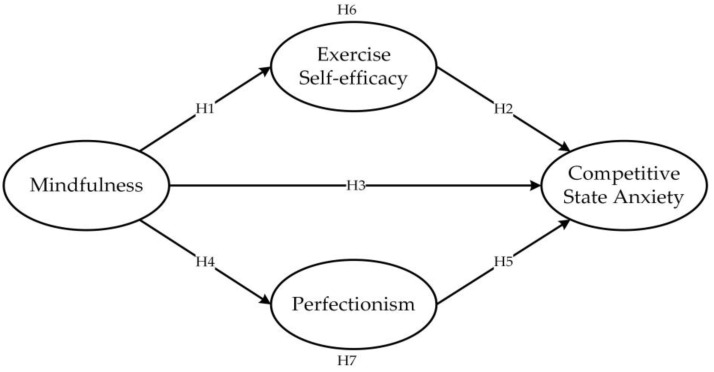
The hypothesized model.

**Figure 2 healthcare-11-02703-f002:**
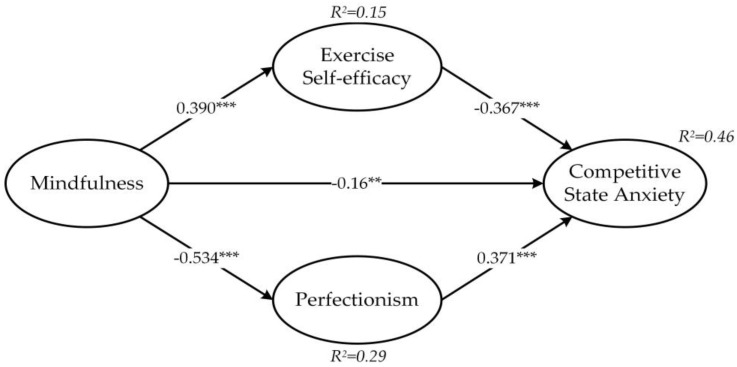
Structural path model. ** *p* < 0.01, *** *p* < 0.001; standardized coefficients are reported.

**Table 1 healthcare-11-02703-t001:** Participant profile (N = 359).

Title	Profiles	Survey (%)
Age	18–20	33 (9.1%)
	21–25	93 (25.9%)
	26–30	122 (34%)
	≥30	111 (31%)
Gender	Male	234 (65.2%)
	Female	125 (34.8%)
Sports items	Ball sports	196 (54.6%)
	Athletics projects	134 (37.3%)
	Other sports	29 (8.1%)
Sports level	Second-level athlete	294 (81.9%)
	Tier 1 athlete	53 (14.8%)
	Athletes at the gym level	12 (3.3%)

**Table 2 healthcare-11-02703-t002:** Reliability and validity tests.

Items	Loadings	Cα	AVE	CR
*Mindfulness (MI)*		0.923	0.707	0.923
MI1	0.801			
MI2	0.845			
MI3	0.861			
MI4	0.854			
MI5	0.842			
*Exercise Self-Efficacy (ESE)*		0.948	0.788	0.949
ESE1	0.829			
ESE2	0.927			
ESE3	0.891			
ESE4	0.919			
ESE5	0.869			
*Perfectionism (PE)*		0.917	0.689	0.917
PE1	0.811			
PE2	0.808			
PE3	0.892			
PE4	0.823			
PE5	0.814			
*Competitive State Anxiety (CSA)*		0.935	0.744	0.935
CSA1	0.869			
CSA2	0.814			
CSA3	0.839			
CSA4	0.898			
CSA5	0.889			

Note: Cα = Cronbach’s alpha; AVE = average variance extracted; CR = composite reliability.

**Table 3 healthcare-11-02703-t003:** Discriminant validity test.

	MI	ESE	PE	CSA
MI	**0.841**			
ESE	0.376 **	**0.888**		
PE	−0.492 **	−0.112 *	**0.830**	
CSA	−0.476 **	−0.461 **	0.477 **	**0.863**

Note: The square root of the average variance extracted (AVE) is in the diagonals (bold); off diagonals is a Pearson’s correlation of contracts. ** *p* < 0.01; * *p* < 0.1. MI = mindfulness; ESE = exercise self-efficacy; PE = perfectionism; CSA = competitive state anxiety.

**Table 4 healthcare-11-02703-t004:** Standardized indirect effects.

	Point Estimation	Product of Coefficients		Bootstrapping				
				Bias-Corrected		Percentile		Two-Tailed Significance
		SE	Z	Lower	Upper	Lower	Upper	
MI → CSA	−0.341	0.056	−6.089	−0.463	−0.240	−0.457	−0.236	0.000 (***)

Note: Standardized estimations of 5000 bootstrap samples. *** *p* < 0.001. MI = mindfulness; CSA = competitive state anxiety.

## Data Availability

The data used to support the findings of this study are available from the corresponding author upon request.

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
