# Peer review of "Association of Mindfulness with Perfectionism, Exercise Self-Efficacy, and Competitive State Anxiety in Injured Athletes Returning to Sports"

_healthcare, 2023, doi:10.3390/healthcare11202703_

Round 1

Reviewer 1 Report

Thank you for submitting to Healthcare.

Unfortunately, the methods and design for the title of this study are very lacking.

As the title suggests, experimental research is appropriate. So, as the title suggests, the author divides the groups into at least two or three groups for verification and makes comparisons after intervening for a certain period of time. Unfortunately, I decided to reject.

Abstract: The boundary between results and conclusions is ambiguous. So add “In conclusion” to indicate the beginning of the conclusion.

Introduction: Lines 67 to 102 are too long. I think you can delete 20-30% of this part.

"2. Literature Review and Hypothesis Development" is not required. Combine this with the introduction.

method

-Explain the process of obtaining the sample size.

table

Table 1: Participants are injured players. There is no information related to the damage, such as what disease it was, how long it was treated, and whether it had surgery.

Table 2: Aren't you using a tool that has already been validated for reliability and validity? Why did reliability and validity tests be conducted in this study?

Table 3: Correlation between methods is not a necessary statistic in the development of a study.

Regarding English, I have no special comments.

This study has more flaws in study design than in English.

Reviewer 2 Report

-I found this to be very comprehensive with mine suggestions:

Lines

12-13: Repeated use " Influence" change to another term

36: In every case ( perhaps add evidence or source that sties every athlete has anxiety when performing. May rectify by stating "May" have 

52: VG definitions and term explanations 

103: VG Organization

119-130: Enjoyed this section as a foundational underpinning of social learning and Bandura. 

228: May define further to reader "Snowball" and "Convenience "

265: Data analysis might be added after results section or within results section... to flow with overall findings

329: VG final development to link your findings and conclusions. Limits are well described and offer insight into further research

Reviewer 3 Report

This manuscript is very clear, relevant and was presented in a well-structured. From the point of view of methodological rigor, the problem of the study was identified, referring its objectives, and presented scientific evidence on the subject of the study. The research variables are defined and the procedures and instruments for the research are well explained. The results are presented, being complemented with a rich discussion. Reflections on the strengths, limitations of the study and future research are carried out. The conclusions are very enlightening and the references are adjusted to the topic under discussion.

In Discussion chapter, line 358, the authors states that “this study is the first to discuss the relationship between exercise self-efficacy and competitive state anxiety among recovering athletes”. Could be there other studies that authors don´t find, so it is suggested to change to “this study discuss the relationship between exercise self-efficacy and competitive state anxiety among recovering athletes”.

Round 2

Reviewer 1 Report

It is positive that there have been many improvements, but some additional revisions are needed.

Add the full word for the abbreviation in the footnotes of the table.

There is no need for the 4.1 4.2 4.3 titles in the discussion.

I have no further comments.

Author Response

Response to Reviewer's Comments - Second Round of Revision

Dear reviewer,

We appreciate the thorough review and insightful comments provided during the second round of evaluation. Your constructive feedback has been instrumental in refining our manuscript, and we are grateful for the opportunity to address the concerns raised. In this response, we outline the revisions made in accordance with your suggestions and provide additional clarification where needed. We are confident that these changes have strengthened the overall quality and contribution of our work.

(1) Add the full word for the abbreviation in the footnotes of the table.

Response 1: Thank you for your suggestions. We have made the necessary revisions, as outlined in the manuscript.

(2) There is no need for the 4.1 4.2 4.3 titles in the discussion.

Response 2: Thank you for your suggestions. We have made the necessary revisions, as outlined in the manuscript.

Once again, thank you for your constructive feedback.